# Institutional Differences and the Choice of Outward Foreign Direct Investment Mode under the "Belt and Road" Initiative: Experience Analysis Based on China's Manufacturing Enterprises

Qing Xie [ID] and Hua Yin *[ID]

School of Business, Central South University, Changsha 410083, China
* Correspondence: hnyinhua@csu.edu.cn

**Abstract:** Based on 1692 outward foreign direct investment (OFDI) events of 735 A-share listed companies in China's manufacturing industry from 2010 to 2019, this paper empirically examines the effect of investment motivation and the impact of institutional differences between China and the host country on the choice of OFDI entry mode; the paper also investigates the moderating effect of the "Belt and Road" Initiative (BRI) on Chinese manufacturing enterprises (CMEs) through use of the logit model. The empirical results show that, with greater institutional differences, CMEs become more inclined to choose cross-border mergers and acquisitions (M&A). Furthermore, a positive moderating effect of resource-seeking motivation on the choice of M&A OFDI by CMEs is observed. The signing of the "Belt and Road" cooperation document positively moderates institutional differences in promoting CMEs—especially state-owned CMEs—to choose the M&A mode. The "Belt and Road" Initiative provides an efficient supply system for OFDI by CMEs. This study enriches and extends existing institutional theories and provides suggestions for the promotion of the geopolitical pattern and international cooperation regarding the "Belt and Road" Initiative.

**Keywords:** outward foreign direct investment; the "Belt and Road" Initiative; institutional differences; cross-border mergers and acquisitions; investment motivation



## 1. Introduction

Institutional supply drives a country's economic growth and is the foundation for encouraging business innovation and sustainable development. A socialist political system with Chinese characteristics has developed in China, and many have benefited from the advantages of this system. China has promoted great development in recent decades, and its comprehensive national strength and international status have significantly improved. From once being backwards to now becoming the world's largest emerging economy, China's economy has become an important engine and stabilizer for world economic growth. The "Belt and Road" Initiative, the construction of the Asian Infrastructure Investment Bank, and China's experience, solutions, and power have been highly recognized by countries worldwide. Although China's political system presents unique advantages and vitality and although the business environment has gradually improved, from a micro-level perspective, there are still gaps between the competitiveness and technological innovation capabilities of Chinese multi-national companies and those of enterprises in developed countries. As such, the government has formulated a series of policy measures to help multi-national companies participate in international competition and cooperation and accelerate their international business operations; examples of these measures are the "going out" strategy and the "Belt and Road" Initiative. Some studies have found that, with the support of national policies and systems in China, OFDI has become one of the most effective and fastest ways for multi-national companies to obtain resources such as markets, technology, and talent [1–6].

The government has attached great importance to OFDI by multi-national companies, supporting their "going out" and deepening their mutually beneficial cooperation with overseas resources. The COVID-19 pandemic has led to a global economic recession, and the global economy has entered a "new normal". International investment has slowed down, growth momentum is lacking, global politics are in turmoil, and international relations and the environmental situation are more complex than ever. Despite the turbulence in the international situation and the many challenges facing the world, China's comprehensive national strength is still growing. The "Belt and Road" Initiative has been essential in promoting global economic recovery and sustainable development, demonstrating China's unique institutional advantages. In 2022, China's OFDI in all industries was USD 146.5 billion, which was an increase of 0.9% compared with the previous year. Chinese domestic investors made non-financial direct investments into 6430 enterprises in 160 countries overseas. Manufacturing is the foundation and basis of a country's strength, and General Secretary Xi Jinping has emphasized that "manufacturing is the lifeline of the national economy." China's manufacturing scale has been the largest in the world for many years, having an irreplaceable role in driving economic development and participating in international competition. However, due to institutional differences, such as political stability, government efficiency, legal system level, market supervision, anti-corruption governance, and property rights protection, between the host country and China, there is a certain degree of uncertainty in international cooperation, which poses risks to OFDI by Chinese companies. In addition, with the impact of the COVID-19 pandemic, many countries are tightening their scrutiny of foreign investment and imposing restrictions on technology transfer, expanding the scope of review and adding new review content, thus significantly increasing restrictions on critical technological fields. Therefore, it is important to understand how to effectively utilize China's institutional and policy advantages, overcome the external institutional constraints on OFDI by Chinese enterprises, reduce external resistance, and improve the performance of overseas investment and operation as these are key parts of the crucial strategic decisions that allow CMEs to conduct business overseas.

The academic community has conducted a series of discussions on the entry mode choice regarding OFDI. Through a literature review, it was found that some authors stressed that, when studying the issue of OFDI, attention should be paid to distinguishing the entry modes of OFDI. Enterprises usually adopt the mode of M&A or greenfield investment to enter overseas markets [7]. Generally, M&A and greenfield investments refer to multi-national enterprises acquiring the assets and equity of existing foreign enterprises or establishing new enterprises in the host country, respectively. Multi-national enterprises can obtain resources from the acquired company through cross-border mergers, such as technology, research and development, and brand. At the same time, greenfield investments enable multi-national enterprises to obtain higher control over the resources and technological knowledge of the newly established company, but also require them to bear fixed investment costs. If multi-national enterprises choose to enter through the M&A approach, the host country must have an efficient financial and equity trading market in order to reduce information asymmetry and market failure, further reducing the operating costs of multi-national enterprises. Furthermore, many studies have discussed the factors that influence OFDI. Considering the influencing factors at the macro-level, it has been stated that, due to the restrictions of country-specific factors, national systems, economic policy uncertainty, national policy risk, cultural distance, national corruption, global competitiveness, the law and order situation, and other factors, there are differences between different OFDI entry mode choices [8–19]. Influential factors at the micro-enterprise behavior level include the firm's resource heterogeneity, strategic orientation, international experience, internal uncertainty, investor sensitivity, and other aspects [20–28]. Among these factors, institutions are an essential driving force for the OFDI entry mode decision making of enterprises. From the perspective of the host country's institutional environment, the higher the quality of the host country's institutional system, the better the host country's policy system and the more inclined the enterprise is to adopt M&A [29]. When the enterprise enters a host

country with a higher institutional quality than the home country, it can formulate effective acquisition strategies by anticipating relevant issues in the acquisition process [30]. The institutional quality has a certain degree of impact on the capital inflow, and the magnitude of the institutional quality effect depends on the stage of economic development [31]. Furthermore, enterprise innovation will be affected by institutional differences and the OFDI mode choice [32]. The critical influence of the differences in institutional regulation, norms, and cognition between the home country and the host country on an enterprise's OFDI entry mode decision-making process cannot be ignored [33]. Although institutional differences will affect the OFDI entry mode choice of an enterprise, when enterprises make investment decisions, national policies can compensate for institutional deficiencies [34].

Since the proposal of the "Belt and Road" Initiative in 2013, it has become not only an essential blueprint for China's current foreign economic construction, but also an essential part of the global governance system. As an institutional innovation, the "Belt and Road" Initiative can compensate for the inadequacy of the international institutional supply, ease the potential friction between China and the countries concerned regarding institutional arrangements, and create platforms and opportunities for Chinese companies to lay out their OFDI space. The countries along the "Belt and Road" are the preferred destinations for outbound investment by enterprises; for example, in 2022, 71.8% of enterprises prioritized countries along the "Belt and Road." The non-financial direct investment of Chinese enterprises in the countries along the "Belt and Road" reached USD 19.16 billion, with a year-on-year increase of 6.5%. The "Belt and Road" Initiative proposed by China is a high-level international political and economic cooperation, and the economic and trade exchanges between China and the countries participating in the "Belt and Road" construction are inseparable. Cooperation partners have gradually expanded from the countries along the "Belt and Road" to 150 countries and 32 international organizations worldwide as of the end of 2022, signing more than 200 co-construction agreements for the "Belt and Road". The countries that have signed "Belt and Road" cooperation agreements with China have policy advantages in economic and trade cooperation. At the same time, the BRI has attracted widespread attention from international and domestic scholars. Based on research on China's OFDI behavior in the countries along the "Belt and Road" [35–40], it has been found that, compared with the quality of the host country's system itself, the systematic difference between China and the countries along the "Belt and Road" has a greater impact on the OFDI behavior and decision preferences of Chinese enterprises. In the process of enterprise internationalization, the choice of the OFDI entry mode by enterprises has become an indispensable part of their OFDI decision-making behavior, and whether the choice is reasonable ultimately affects the investment and operational performance of enterprises. Based on previous research on the relationship between institutional quality and enterprise OFDI behavior, this article focuses on analyzing the impact of institutional differences on the choice of OFDI entry mode by Chinese enterprises as well as analyzing the "Belt and Road" construction in detail.

In recent years, papers on OFDI and its influencing factors, as well as the relationships between the BRI and OFDI, have proliferated; however, there remain various shortcomings, such as those detailed in the following. First, the existing literature has generally analyzed the factors influencing the OFDI entry mode, including institutional factors. These studies limited the relationship between the quality of the host country or home country's institutional system and the OFDI investment mode of enterprises to a unilateral institutional perspective, ignoring the impact of bilateral institutional differences on OFDI mode choice and the role of national policies. As such, the literature on the different entry modes of Chinese multi-national firms lacks a systematic, comprehensive explanation and a unified analytical framework from an institutional perspective to discuss different outcomes. Second, previous studies have discussed the investment motivations for OFDI by Chinese enterprises, providing reference significance for exploring the role of heterogeneous investment motivations in the choice of OFDI entry mode by CMEs. However, the existing literature still lacks research results from the perspective of heterogeneous investment

motivations in order to explore the relationship between the institutional environment and the choice of OFDI mode, thus failing to better grasp the regularity of the "heterogeneous preferences" of OFDI by CMEs in the dynamic evolution process of institutions. Finally, when Chinese multi-national companies actively integrate into the international market, the Chinese government strongly supports their "going out" strategy. However, there remains a lack of extensive empirical research on how government policies affect different OFDI entry modes. Therefore, this article attempts to combine the characteristics of China's BRI era through the incorporation of the expanded circle of "Belt and Road" partners into the research content and further explores the moderating effect of the initiative in the relationship between institutional differences and the OFDI entry mode choice of CMEs.

Previous reports have mostly studied the OFDI entry mode choice of enterprises based on the institutions of the host and home countries. Indeed, many factors, including the pressure from the host country institutional environment, home country institutional environment, and institutional differences between the target country and the home country, should be explored before the enterprise enters into the target country. However, few studies have paid attention to the impact of institutional differences on the choice of enterprise OFDI mode and even more rarely in the context of developing economies. Based on the characteristics of developing economies, this paper discusses the specific situation of the OFDI entry modes of CMEs, enriching and expanding institutional theories of enterprise OFDI behavior research. Furthermore, our examination details a beneficial exploration of embedding BRI and investment motivation situation factors. Based on the micro-data of Chinese manufacturing enterprises, this paper deeply analyzes the strategic decision from two dimensions of the BRI—including the national public policy and the enterprise's different investment motivations—providing powerful micro-evidence for enriching research results regarding the "Belt and Road" construction. Our results also highlight the way in which Chinese enterprises can choose an appropriate OFDI entry mode when facing different institutional environments in target countries, which is expected to be conducive to reducing the failure rate of investment cooperation projects caused by the improper decisions of Chinese enterprises in the internationalization process. Collectively, this study has important academic, theoretical, and practical value in boosting cooperation between Chinese enterprises and those in the "Belt and Road" countries.

The novelty of this article lies in its adoption of multiple perspectives to thoroughly and systematically examine the impact of institutional differences in the Chinese context. The BRI includes a wide range of countries and regions spanning the Asian, European, and African continents. There are certain complexities and differences in the economic development stage and institutional rules of various countries. Their diversified characteristics bring a series of challenges to the OFDI behavior decision making of Chinese enterprises. We fully consider international investment cooperation, with the BRI supply making up for the international legal system and transnational trading rules. Based on the common behavior standards and rules to strengthen cooperation, the signing of the "Belt and Road" cooperation document has played an important role in the decision making of CMEs with respect to their outgoing OFDI behavior. Based on different cooperation needs, the target country should be effectively evaluated such that differentiated policy recommendations can be made for the OFDI. Specifically, the main purpose of this paper is to use empirical analysis and literature evidence in order to integrate the choice of OFDI entry mode into the analytical framework of CMEs in the construction of BRI from the perspective of institutional differences. Through understanding the relationship and differences between the institution of the target country and China's institution, the OFDI entry mode may be effectively altered. Panel data are used to assess the relevant content of China's OFDI events, allowing us to conduct a detailed empirical analysis and provide a relevant discussion.

The main contributions of this article are as follows: First, unlike previous studies that have only focused on either the home country or host country's institutional environment's impact on the choice of OFDI entry mode by Chinese enterprises, this article

incorporates the impact of institutional differences on the OFDI entry mode choice into the framework of the BRI's geopolitical pattern and international cooperation, which has a distinct contemporary significance and provides practical theoretical and empirical evidence to effectively promote the construction of the BRI. Second, from the perspectives of investment motivation and policy advantages, this article explores the moderating effects of these two factors on the OFDI mode choices of CMEs. Compared with previous literature discussing countries along the "Belt and Road," this study focuses on countries that have signed cooperation agreements with China under the Initiative, revealing the expanding "Belt and Road" partners and presenting a certain degree of dynamism. Third, this study attempts to introduce the political affinity index from the United Nations General Assembly voting data as a tool variable in order to identify the impact of institutional differences on the OFDI entry mode choices of Chinese enterprises and the moderating effects of the "Belt and Road" Initiative. Fourth, by combining macro- and micro-level data, this study comprehensively considers micro-level data on the OFDI of enterprises that is matched with multiple databases, annual reports, and macro-level data at the national level. This not only provides theoretical guidance for the investment decisions of Chinese enterprises in countries along the "Belt and Road," but also helps and promotes the construction of infrastructure and economic growth in these countries while also providing a "Chinese solution" for international investment cooperation in participating countries along the BRI. In summary, this paper aims to make contributions that improve the competitiveness of CMEs in the international market by seeking common interests with "Belt and Road" partners to the greatest extent and by promoting the economic development of partner countries and the high-quality development of China's manufacturing industry.

The theoretical framework of this paper is based on the theoretical explanation of how institutional differences affect the choice of OFDI entry mode, which form a hypothesis to be tested. Second, the direct effect of institutional differences on the choice of OFDI mode by enterprises is examined, and the mechanism formed according to different quality dimensions is further discussed. Third, the moderating effects of different investment motivations and the BRI are examined. Then, heterogeneity analyses are conducted at the enterprise ownership structure and national regional levels. Finally, by identifying instrumental variables, an endogeneity test is conducted.

## 2. Theoretical Analysis and Research Hypothesis

Institutional differences refer to the similarities and differences in the quality of institutions (e.g., political stability and legal rules) between two countries or regions. An institution refers to the organic integration of social governance and political operation rules that are closely related to the operation of the market economy, which has an essential impact on the operation of the market economy and transaction costs in society. Institutions can be considered as a set of game rules and constraints in society, where it is vital to understand the relationship between politics and the economy as well as how this relationship affects economic growth [41]. Institutions consist of elements and resources that provide regulatory, cognitive, and stability functions for social life [42]. From national constitutions to specific internal regulations, as well as national activities to individual behavior, they are all included in the formal constraints of institutions. The basic function of an institution is to effectively exert the market mechanism and reduce the transaction costs or risks of enterprises and individuals in the market [43].

National institutions have received widespread attention due to their essential role in economic and social behavior and development [44]; however, the international social environment is complex and constantly changing, and there are institutional differences between countries of different social forms. The differences in national institutions are essential factors that affect a country's OFDI strategic choices and are critical factors that affect the motivation and ability of multi-national enterprises to carry out OFDI. McMillan [45] has pointed out that neglecting institutional factors becomes particularly evident when the market cannot function normally. When using mainstream theories to explain

investment in developing countries, treating institutional factors as exogenous variables can pose significant challenges to research [46]. Formal and informal institutional factors cannot simply be regarded as "background," as they have become variables that influence organizational investment decisions. The investment decisions of enterprises result from an interaction between institutions and organizations [43,47]. As such, institutions are not just background conditions; they are endogenous variables affecting how enterprises plan and implement their strategic choices and create competitive advantages [48].

When multi-national corporations engage in OFDI activities, the influence of the host country's institutions is not just reflected in economic aspects, such as the business environment [49]. In terms of institutional quality, there are also "natural" differences between the national institutions of the home and host countries. If the host country has an ideal institutional environment, it can provide information about business partners and their possible behaviors, reduce information asymmetry and market failure, and thus reduce business costs. However, if the institutional environment of the host country is relatively poor, it will increase information asymmetry, causing Chinese enterprises to spend more resources in the search for information and increasing business uncertainty. Therefore, enterprises choose organizational forms to reduce transaction costs based on the institutional environment, and different organizational forms may bring differentiated transaction costs. In a poor institutional environment, the host country's government may also set up various barriers to entry, thereby increasing the possibility of Chinese enterprises choosing the M&A approach. If entering through M&A, the host country must have efficient market mechanisms, primarily, financial and equity trading markets. Effective market institutions ensure information symmetry, predictability, and the effective implementation of financial market transactions; however, in a poor institutional environment, the securities market lacks liquidity, effective financial intermediaries, and market volatility, thus reducing potential M&A transactions. As greenfield investment does not involve equity and asset transactions, Chinese enterprises may also adopt the greenfield investment approach.

The World Bank began publishing the "World Governance Indicators" (WGI) in 1996. This index divides institutional quality into six dimensions: political stability, government effectiveness, regulatory quality, rule of law, voice and accountability, and control of corruption. When examining the institutional differences between China and the host country, the host country can be divided into two categories: high institutional quality and low institutional quality. First, for host countries with a high institutional quality, the greater the institutional differences, the more superior their market economy and political system, and multi-national enterprises can use their technological advantages to profit in a well-established corporate control market. If the enterprise adopts the M&A entry mode, the investment efficiency can be improved. Second, for host countries with a low institutional quality, the greater the institutional differences and the more incomplete the legal system of the investment target country. Due to the high investment risks associated with the capital invested in new enterprises, if enterprises choose to enter the host country through M&A, they can reduce the risks associated with setting up new enterprises, save the time needed to train employees, reduce organizational and opportunity costs, and improve profitability. Therefore, enterprises may also prefer to choose M&A in this context [21]. In terms of emerging market countries, Rienda et al. [50] took India as an example. They found that, due to factors such as transaction costs and time costs, Indian companies prefer to enter host countries with large institutional differences through M&A rather than greenfield investments. Ramasamy et al. [51] found that some large enterprises in China prefer to adopt M&A in high political risk countries in response to national macro-development strategies.

The image of "Made in China" has quietly changed in recent years; however, the problem of "big but not strong" still exists in China's manufacturing industry. As a key to progress, China's manufacturing industry is in a critical development period. The international competitiveness of manufacturing enterprises is not strong enough, and it is necessary to accelerate the technological innovation, transformation, and upgrading while

fully utilizing and integrating the existing production capacity and resources within the industry, as well as sharing research and development achievements and sales networks with host countries to realize the interests of both sides; these aspects form a mutually beneficial and advantageous cooperation with respect to production capacity. When CMEs cooperate with countries or regions with significant differences in the institutional environment, M&A can save construction time, improve resource allocation efficiency, and leverage resource advantages to enhance market power, thus obtaining maximized benefits. When CMEs cooperate with countries or regions with relatively small differences in institutional environment, cooperation between Chinese enterprises and the host country companies is smoother due to the similar institutional and business environments. Chinese companies have less dependence on localized resources, face weaker institutional risks, have more robust transferable capabilities, and tend to choose greenfield investment by investing in and building factories in the target investment country to make a profit. Based on this, we propose the following hypothesis:

**Hypothesis 1.** *The greater the institutional differences, the more CMEs choose M&A; the smaller the institutional differences, the more CMEs choose greenfield investment.*

Investment motivation refers to the purpose that the investment activity subject aims to achieve when conducting foreign investment activities. Institutions directly affect the market entry strategy of multi-national enterprises, and this effect is regulated by the demand of the multi-national enterprise for different types of local resources [29]. Based on the investment motivations for OFDI by CMEs, this article mainly studies three types of investment motivations: market-seeking, technology-seeking, and resource-seeking. Market-seeking investment motivation is associated with the horizontal expansion of enterprises. Generally, outward direct investment with a market-seeking motivation intends to open up new overseas markets or maintain them. After the 1990s, the rise of emerging countries increased the scale of OFDI by multi-national companies, opening up international markets. Market-seeking investment motivation is mainly manifested in the following four aspects: understanding the needs of local customers at close range, breaking through multiple institutional monopolies and avoiding trade barriers in the host country, collecting local market information, and enhancing the popularity of independent brands locally. It can be seen that the market-seeking investment motivation tends to weaken the impact of institutional differences through greenfield investment. Technology-seeking motivation aims to obtain technological progress, through which multi-national companies learn about advanced management experience, technical equipment, brands, and distribution networks from the host country. When the institutional differences between the host country and the home country of the multi-national enterprise are slight, multi-national enterprises tend to establish research and development centers or establish overseas companies to optimize the production process configuration through industrial linkage effects, thus promoting specialized production and improving the technical innovation level of the parent company. When the institutional differences between the host country and the home country of the multi-national enterprise are large, based on the technology-seeking motivation, multi-national companies tend to choose M&A.

Moreover, by directly learning and absorbing the acquired company's core production technology and management experience in the host country and rapidly feeding back advanced technology and experience to the parent company through reverse technology spillover effects, the parent company's technological innovation capabilities can be rapidly improved. Resource-seeking OFDI is mainly carried out to fully utilize the rich natural resources and cheap raw materials in a local area, thus breaking the limitations of resource shortages and developing certain industries that cannot be developed or have high development costs due to the shortage of resources in the home country. In the case of resource-seeking, multi-national companies are like "outsiders," who are not only subject to legal restrictions by the government but also receive discriminatory treatment

from the public, making it more challenging to obtain a legitimate status. It follows that resource-seeking OFDI tends to exaggerate the results of institutional differences. Therefore, in countries with more significant institutional differences from the home country, multi-national companies tend to choose M&A for investment. Based on the above analysis, we propose the following hypotheses:

**Hypothesis 2.** *Market-seeking motivation has a negative effect on the relationship between institutional differences and cross-border M&A by CMEs.*

**Hypothesis 3.** *Technology-seeking motivation has a positive effect on the relationship between institutional differences and cross-border M&A by CMEs.*

**Hypothesis 4.** *Resource-seeking motivation has a positive effect on the relationship between institutional differences and cross-border M&A by CMEs.*

Governments of emerging market countries have issued rules and regulations for OFDI, encouraging OFDI projects that promote export-oriented growth strategies and creating favorable conditions for the overseas strategic asset acquisitions of domestic multi-national corporations [52]. In some cases, enterprises may also deploy political resources to actively obtain favorable operating conditions or evade unfavorable regulations through political action. Traditional economic theory assumes that enterprises are passive in response to the institutional environment in which they operate, either complying with regulations or exiting the market. They seek legitimacy within the given institutional framework to avoid costs, compete for resources, and strive for maximum profits; however, some enterprises are capable of and are willing to influence institutional frameworks to evade unfavorable regulations, and this ability to influence the institutional framework is referred to as "institutional capital" [53]. If the market strategy is essential and an enterprise's political resources are powerful enough to influence government policies, the enterprise can and will negotiate with the government. The benefits of doing so are numerous, such as obtaining unique resources and protection, improving the enterprise's market position, or reducing costs. However, the most crucial benefit is the legitimacy of the enterprise's business behavior, which increases opportunities for survival and profit margins.

Due to the different national conditions and development stages of the countries involved in the BRI, their sustainable development paths differ [54]. When Chinese companies enter into "Belt and Road" countries with significant differences in their institutional environments, they are affected by factors such as the speed of the overseas market entry, the degree of access to proprietary resources, and the degree of uncertainty. To reduce transaction costs and obtain more local resources, Chinese enterprises are more willing to directly acquire companies in these countries and carry out reasonable resource allocation and effective integration [55]. National policies can effectively compensate for the institutional gaps in the host country and the imbalance of the home country's institutional environment. This compensatory effect is even greater for host countries with poor institutional environments. At the same time, the government provides policy support for corporate internationalization in order to protect the interests associated with outward investment by enterprises [56]. The "Belt and Road" cooperation documents are landmark documents that establish a consensus between the involved countries, reflecting the process of policy communication between the two sides, protection of the rights and interests of both parties, and a willingness to coordinate and strengthen bilateral relations under a priority framework. The more essential documents signed between the host country and China, the deeper the policy communication and strategic cooperation between the two countries. The "Belt and Road" Initiative has created a favorable external environment for CMEs to integrate into international production capacity cooperation as a critical industry in international production capacity cooperation. Due to the incentive of "policy tasks,"

enterprises may implement national policies by adopting M&A. Based on this, we propose the following hypothesis:

**Hypothesis 5.** *If the investment destination country signs a "Belt and Road" cooperation document with China, institutional differences will prompt CMEs to tend to choose M&A.*

## 3. Research Design

### 3.1. Data Sources and Sample Selection

Due to the severe impact of the COVID-19 pandemic on global investment, we selected 1692 samples of OFDI events (718 cross-border mergers and acquisitions and 974 greenfield investments) by 735 CMEs listed as Shanghai and Shenzhen A-share companies from 2010 to 2019. To avoid the influence of previous OFDIs, the sample only includes the company's first OFDI event in the target country. The data were mainly obtained from the Public Directory of Filing Results of Overseas Investment Enterprises (Institutions) [57], Belt and Road Portal [58], World Bank WDI database, fDi Markets database, CNRDS China Research Data Service Platform, the Guotaian database, and the annual reports of the listed companies.

#### 3.1.1. Dependent Variable

The dependent variable in this study is whether CMEs conduct investment in destination countries through M&A (*MA*). A value of 1 is assigned if the firm follows the *MA* mode; otherwise, it is valued 0.

#### 3.1.2. Independent Variable

Regarding institutional differences, we use the Worldwide Governance Indicators (WGI) to measure the quality of institutions in different countries. Following previous studies [59–61], the average score of the six dimensions of WGI is calculated to obtain the institutional quality score of each country, from which the score of China's institutional quality is subtracted to obtain the institutional difference between the two countries.

#### 3.1.3. Control Variables

According to previous research [62–64], we controlled relevant variables including enterprise size, enterprise fixed assets, enterprise capital intensity, enterprise age, infrastructure construction, and geographical distance.

#### 3.1.4. Moderating Variables

We divide the motivations for enterprise investment into three categories—market-seeking, technology-seeking, and resource-seeking—and use these three types of investment motivations as moderating variables. The interaction term between institutional differences and investment motivation is constructed in order to further examine the moderating effect of enterprise investment motivation. We also take the signing of the "Belt and Road" cooperation document as a moderating variable and construct an interaction term between institutional differences and the signing of the "Belt and Road" cooperation document to further examine the moderating effect of the BRI.

#### 3.1.5. Instrumental Variable

We take the political affinity index in the voting data of the United Nations General Assembly to measure the similarity of the institutions of the two countries. The political affinity index between any two countries is calculated based on the three-category method of voting behavior ("for", "abstention", and "against") and is used as an instrumental variable for 2SLS regression. The political affinity index is calculated using the method of Signorino and Ritter [65], with values ranging from −1 to 1. The larger the value, the more similar that the political stance and preferences of the two countries are, and the higher

the strategic consensus formed in critical international affairs of the United Nations. The meanings of variables are provided in Table 1.

**Table 1.** Variables and measurements.

| Variable | Variable Name | Variable Measurement |
|---|---|---|
| Dependent variable | *MA* | The value of M&A is 1, and the value of greenfield investment is 0 |
| Independent variable | *idiff* | Institutional differences, differences in the WGI index between China and the host country |
| Moderating variables | *gdp* | Market-seeking, GDP of the host country |
| | *tec* | Technology-seeking, the proportion of high-tech products exports |
| | *res* | Resource-seeking, the proportion of metal and mineral exports |
| | *cd* | The host country signs the "Belt and Road" cooperation agreement with China; *cd* takes the value of 1 and 0 otherwise. |
| Control variables | *lnstaff* | Enterprise size, natural log of the number of workers |
| | *lnfa* | Enterprise fixed assets, the natural log of the net value of fixed assets |
| | *cap* | Enterprise capital intensity, the ratio of total assets to operating income |
| | *age* | Enterprise age, the time interval between the establishment of the firm and the occurrence of the investment event |
| | *lngfcf* | Infrastructure construction, the natural log of the capital formation to GDP ratio |
| | *lndis* | Geographical distance, natural log of the straight-line distance between Beijing and the capital of the host country |
| Instrumental variable | *ipd* | The political affinity index, based on the three-category method of voting behavior between any two countries, is calculated |

### 3.2. Empirical Model

Because the dependent variable in this study is binary and cannot take continuous values, and because there are multiple binary variables in the independent and control variables, we use the logit model. Stata15 software (StataCorp LLC, Texas, USA) was used for the empirical analysis, and the following regression model equation was constructed:

$$Logit\left[P\left(MA_{ijt} = 1\right)\right] = \alpha_0 + \alpha_1 idiff_{ijt} + \alpha_2 \sum \theta_n X_n + \lambda_i + \nu_t + \varepsilon_{ijt}, \tag{1}$$

where *i*, *j*, and *t* index the firm, country, and time, respectively. In this paper, whether the firm adopts the *MA* mode to enter the host country is taken as the dependent variable to be explained, and $P(MA_{ijt} = 1)$ represents that the firm enters the host country and chooses M&A (*MA* = 1 if acquired; otherwise, it is 0). Furthermore, $idiff_{ijt}$ represents the institutional differences between China and host country *j*; $X_n$ denotes the set of control variable; $\lambda_i$ and $\nu_t$ represent the province and time fixed-effects, respectively; and $\varepsilon_{ijt}$ denotes the model error term.

We consider the role of firm investment motivation in the choice of OFDI entry mode for CMEs in the context of institutional differences between China and the host country. Regression tests were conducted with respect to investment motivations, including technology-seeking, resource-seeking, and market-seeking motivations, in order to examine the moderating effect of firm investment motivation. The specific model was set as follows:

$$Logit\left[P\left(MA_{ijt} = 1\right)\right] = \varphi_0 + \varphi_1 idiff_{ijt} + \varphi_2 Moti_{ijt} + \varphi_3 idiff_{ijt} \times Moti_{ijt} + \sum \varphi_n X_n + \lambda_i + \nu_t + \varepsilon_{ijt}, \tag{2}$$

where *Moti* represents the investment motivation of the enterprise. We constructed the interaction term *idiff\*Moti* between the institutional differences and enterprise investment motivations (where the interaction term between the institutional differences and market-seeking motivation is *idiff\*gdp*, the interaction term between institutional differences and technology-seeking motivation is *idiff\*tec*, and the interaction term between institutional differences and resource-seeking motivation is *idiff\*res*). The GDP of the host country is used to measure market-seeking motivation, the proportion of high-tech product exports in the host country is used to measure technology-seeking motivation, and the proportion of metal and mineral exports in the host country is used to measure resource-seeking motivation. The data on investment motivations were all sourced from the World Bank WDI database.

To further consider the moderating role of the "Belt and Road" Initiative on the impact of institutional differences on the OFDI entry mode choice of CMEs, the signing of "Belt and Road" cooperation agreements between the host country and China was taken as a moderating variable for the regression analysis in order to examine the moderating effect of the BRI. The specific model was set as follows:

$$Logit\big[P\big(MA_{ijt}=1\big)\big] = \beta_0 + \beta_1 idiff_{ijt} + \beta_2 cd_{ijt} + \beta_3 idiff_{ijt} \times cd_{ijt} + \sum \beta_n X_n + \lambda_i + \nu_t + \varepsilon_{ijt}, \tag{3}$$

where *cd* represents the signing of the "Belt and Road" cooperation agreement between the host country and China; if the host country had signed the "Belt and Road" cooperation agreement with China before the OFDI event, *cd* takes a value of 1; otherwise, it takes a value of 0. To further investigate the moderating effect of the BRI, an interaction term *idiff\*cd* was constructed between the institutional difference and the signing of the "Belt and Road" cooperation agreement. To avoid multicollinearity, the institutional difference, investment motivation, and other variables were centralized when setting the interaction terms in this paper.

## 4. Empirical Results and Analysis

### 4.1. Descriptive Statistics

The descriptive statistics for the variables used in this study are presented in Table 2. The mean, standard deviation, minimum, and maximum values for the institutional difference (*idiff*) between China and the host countries were 1.2252, 0.8335, −0.9066, and 2.3884, respectively, indicating a significant difference in the quality of institutions with respect to political stability, government efficiency, and other factors. Some countries presented high institutional quality, while others had low institutional quality, warranting further investigation of the impact of this difference on the OFDI entry mode choice of CMEs. A maximum likelihood estimation analysis was used to determine the regression coefficients, and multicollinearity was tested for using the variance inflation factor (VIF). The results indicate that the VIF value for the primary independent variable—institutional difference—was 1.13, while the maximum VIF value for the other variables was 4.20. Thus, all variables had VIF values did not exceed the critical value of 10 for multicollinearity, indicating that the selected model did not suffer from multicollinearity.

**Table 2.** Descriptive statistics of main variables.

| Variables | Samples | Mean | Std. | Max | Min | VIF |
|-----------|---------|--------|--------|---------|---------|------|
| *MA* | 1692 | 0.4243 | 0.4944 | 0.0000 | 1.0000 | |
| *idiff* | 1692 | 1.2252 | 0.8335 | −0.9066 | 2.3884 | 1.13 |
| *lnstaff* | 1692 | 8.4773 | 1.4796 | 4.0604 | 12.3422 | 4.20 |
| *lnfa* | 1692 | 20.9621 | 1.6889 | 15.8164 | 25.7617 | 3.85 |
| *cap* | 1692 | 2.0163 | 1.2634 | 0.2768 | 15.6189 | 1.23 |
| *age* | 1692 | 16.7340 | 5.7744 | 2.0000 | 49.0000 | 1.07 |
| *lngfcf* | 1692 | 3.1262 | 0.1962 | 2.2721 | 3.9275 | 1.24 |
| *lndis* | 1692 | 8.7120 | 0.6609 | 6.8624 | 9.8677 | 1.21 |

To test the hypotheses regarding the impact of institutional differences on the OFDI entry mode choice of CMEs, we conducted a correlation analysis of the main variables affecting the OFDI entry mode. The results are shown in Table 3, and all correlation coefficients were below the critical value of one.

**Table 3.** Descriptive statistics: correlation coefficients.

| Variables | 1 | 2 | 3 | 4 | 5 | 6 | 7 | 8 |
|---|---|---|---|---|---|---|---|---|
| *MA* | 1.000 | | | | | | | |
| *idiff* | 0.386 *** | 1.000 | | | | | | |
| *lnstaff* | −0.130 *** | −0.213 *** | 1.000 | | | | | |
| *lnfa* | −0.076 *** | −0.223 *** | 0.647 *** | 1.000 | | | | |
| *cap* | 0.118 *** | 0.090 *** | −0.367 *** | −0.190 *** | 1.000 | | | |
| *age* | −0.018 | −0.189 *** | 0.194 *** | 0.198 *** | −0.085 *** | 1.000 | | |
| *lngfcf* | −0.161 *** | −0.194 *** | 0.007 | 0.015 | −0.006 | −0.019 | 1.000 | |
| *lndis* | 0.114 *** | −0.002 | 0.076 *** | 0.126 *** | 0.040 | 0.067 ** | −0.387 *** | 1.000 |

Note: ** and *** represent significance at the 5%, and 1% levels, respectively.

### 4.2. Baseline Regression Analysis

Table 4 shows the basic regression results for the effect of institutional differences on the OFDI entry mode choice of CMEs. According to columns (1)–(4), as the control variables were gradually added, and as province and time-fixed effects were introduced, the regression coefficient of institutional differences remained consistently positive and significant at the 1% level. This suggests that the greater the institutional differences between China and the host country, the more likely that CMEs are to choose M&A as their investment mode, while smaller institutional differences may lead to the choice of greenfield investment. This conclusion validates Hypothesis 1. A possible explanation is that, if the institutional level of the host country is higher than that of China, greater differences in the institutional environment indicate a complete market economy and political system, which can help Chinese firms utilize local management experience and technology entirely. Therefore, they are more willing to choose the M&A mode when entering the host country market, thus improving their investment efficiency quickly. Suppose the institutional level of the host country is lower than that of China; then, greater differences in the institutional environment suggest more risks and challenges, such as political instability, inefficient government, harsh market competition, and even the possibility of forced termination of cooperation. In this case, M&A can weaken investment uncertainty and reduce transaction and time costs. When cooperating with countries with similar institutional characteristics to China, newly established enterprises are less likely to be restricted by the host country's laws and policies, and they can also fully utilize local market resources to reduce costs. Therefore, the greenfield investment mode can lead to a better utilization of the transferrable advantages of the firm itself and can achieve maximum profit.

Regarding the control variables, the regression coefficients of fixed assets, capital intensity, enterprise age, and geographical distance were all significantly positive while those of enterprise size and host country infrastructure were significantly negative. These results indicate that the selected control variables are important factors influencing the OFDI mode choice of CMEs. When the values of fixed assets, capital intensity, enterprise age, and geographical distance are more significant, CMEs are more likely to choose M&A; meanwhile, when the values of enterprise size and host country infrastructure are larger, CMEs are more likely to choose greenfield investment.

**Table 4.** Results of basic regression.

| | (1) | (2) | (3) | (4) |
|---|---|---|---|---|
| *idiff* | 1.165 *** | 1.170 *** | 1.142 *** | 1.168 *** |
| | (0.078) | (0.082) | (0.080) | (0.084) |
| *lnstaff* | | −0.254 *** | −0.257 *** | −0.246 *** |
| | | (0.084) | (0.078) | (0.085) |
| *lnfa* | | 0.206 *** | 0.191 *** | 0.203 *** |
| | | (0.069) | (0.066) | (0.070) |
| *cap* | | 0.143 ** | 0.086 * | 0.125 ** |
| | | (0.056) | (0.050) | (0.056) |
| *age* | | 0.026 *** | 0.018 * | 0.022 ** |
| | | (0.010) | (0.010) | (0.011) |
| *lngfcf* | | −0.820 ** | −0.849 ** | −0.785 ** |
| | | (0.368) | (0.357) | (0.373) |
| *lndis* | | 0.285 *** | 0.266 *** | 0.272 *** |
| | | (0.097) | (0.096) | (0.097) |
| Constant | −1.264 ** | −4.218 ** | −3.512 * | −3.780 * |
| | (0.492) | (1.877) | (1.829) | (1.945) |
| Province FE | Yes | Yes | No | Yes |
| Year FE | Yes | No | Yes | Yes |
| Observations | 1692 | 1692 | 1692 | 1692 |
| Pseudo $R^2$ | 0.157 | 0.174 | 0.155 | 0.180 |
| Log Lik | −972.3 | −953.0 | −974.9 | −945.9 |
| LR $Chi^2$ | 295.4 | 300.2 | 289.2 | 317.6 |
| Prob > $chi^2$ | 0.000 | 0.000 | 0.000 | 0.000 |

Note: *, **, and *** represent significance at the 10%, 5%, and 1% levels, respectively; the robust standard errors are in parentheses.

### 4.3. Regression Analysis of Different Dimensions of Institutional Quality

The World Bank's WGI, which includes six dimensions of indicators—namely, voice and accountability, political stability, government effectiveness, regulatory quality, rule of law, and control of corruption—have been released since 1996. Based on the significant impact of institutional differences on the OFDI mode choice of CMEs, we further explored the effects of these six dimensions of institutional quality.

Table 5 shows the different effects of the specific dimensions of institutional quality. It can be concluded from (1)-(6) that all six dimensions (i.e., voice and accountability, political stability, government effectiveness, regulatory quality, rule of law, and control of corruption) were significantly positive at the 1% level. This indicates that the influence of all different dimensions of institutional quality on the OFDI mode choice of CMEs was highly significant. The positive effects of these variables indicate that CMEs should pay more attention to and carefully consider institutional differences when investing in target countries. Voice and accountability was the dimension with the highest degree of influence among the six dimensions, which fully demonstrates that the election rights and freedom of expression of local citizens can effectively attract M&A by CMEs as they indirectly reflect the importance of the host country's democratic atmosphere in attracting foreign investment.

**Table 5.** Regression results of institutional quality.

| | (1) | (2) | (3) | (4) | (5) | (6) |
|---|---|---|---|---|---|---|
| Voice and accountability | 1.422 *** | | | | | |
| | (0.116) | | | | | |
| Political stability | | 1.041 *** | | | | |
| | | (0.097) | | | | |

**Table 5.** *Cont.*

| | (1) | (2) | (3) | (4) | (5) | (6) |
|---|---|---|---|---|---|---|
| Government effectiveness | | | 1.027 *** | | | |
| | | | (0.077) | | | |
| Regulatory quality | | | | 0.954 *** | | |
| | | | | (0.072) | | |
| Rule of law | | | | | 0.981 *** | |
| | | | | | (0.072) | |
| Control of corruption | | | | | | 0.850 *** |
| | | | | | | (0.063) |
| *lnstaff* | −0.253 *** | −0.228 *** | −0.255 *** | −0.242 *** | −0.250 *** | −0.263 *** |
| | (0.083) | (0.084) | (0.084) | (0.083) | (0.084) | (0.084) |
| *lnfa* | 0.204 *** | 0.162 ** | 0.199 *** | 0.192 *** | 0.200 *** | 0.202 *** |
| | (0.069) | (0.069) | (0.069) | (0.069) | (0.069) | (0.069) |
| *cap* | 0.129 ** | 0.128 ** | 0.125 ** | 0.127 ** | 0.126 ** | 0.116 ** |
| | (0.058) | (0.056) | (0.054) | (0.055) | (0.056) | (0.054) |
| *age* | 0.019 * | 0.019 * | 0.019 * | 0.021 * | 0.021 * | 0.021 * |
| | (0.011) | (0.010) | (0.011) | (0.011) | (0.011) | (0.011) |
| *lngfcf* | −0.896 *** | −1.123 *** | −0.963 *** | −0.584 | −0.863 ** | −0.970 *** |
| | (0.336) | (0.357) | (0.362) | (0.363) | (0.359) | (0.347) |
| *lndis* | −0.288 *** | 0.328 *** | 0.396 *** | 0.425 *** | 0.298 *** | 0.253 *** |
| | (0.099) | (0.097) | (0.099) | (0.100) | (0.096) | (0.097) |
| Constant | 2.260 | −1.179 | −3.534 * | −4.780 ** | −2.928 | −2.029 |
| | (1.821) | (1.866) | (1.896) | (1.927) | (1.886) | (1.825) |
| Province FE | Yes | Yes | Yes | Yes | Yes | Yes |
| Year FE | Yes | Yes | Yes | Yes | Yes | Yes |
| Observations | 1692 | 1692 | 1692 | 1692 | 1692 | 1692 |
| Pseudo $R^2$ | 0.193 | 0.150 | 0.164 | 0.165 | 0.173 | 0.167 |
| Log Lik | −931.2 | −980.7 | −964.7 | −963.3 | −954.3 | −960.6 |
| LR $Chi^2$ | 266.8 | 244.3 | 310.1 | 302.1 | 320.7 | 313.4 |
| Prob > $chi^2$ | 0.000 | 0.000 | 0.000 | 0.000 | 0.000 | 0.000 |

Note: *, **, and *** represent significance at the 10%, 5%, and 1% levels, respectively; the robust standard errors are in parentheses.

### 4.4. Moderating Effects

Moderating effects of different investment motivations. We analyzed the indirect effects of institutional differences and investment motivations on the choice of OFDI entry mode by CMEs through the introduction of an interaction term between institutional differences and investment motivations. The results are shown in Table 6. According to columns (1)–(2), the interaction term *idiff\*lngdp* between the institutional differences and market-seeking motivation was negative but not significant. The interaction term *idiff\*tec* in columns (3)–(4) between the institutional differences and technology-seeking motivation was negative and significant at the 1% level, indicating that CMEs tend to invest in greenfield investments in countries with smaller institutional differences when engaging in technology-seeking investments. Thus, technology-seeking motivation negatively moderates the relationship between institutional differences and the choice of M&A for OFDI by CMEs, contrary to Hypothesis 3. A possible reason for this may be that, in recent years, Chinese companies have faced continuous turbulence in the international environment, the total amount of China's OFDI and the overseas M&A investment of the Chinese manufacturing industry have been significantly hindered, and the scale of M&A has dramatically decreased. Against this background, the scale of greenfield investments in the host country by CMEs has shown a countertrend of stable growth based on technology-seeking motivations. The threshold for M&A investment with technology-seeking motivation by CMEs has sharply increased, and CMEs have adjusted their investment strategies promptly to better enter overseas markets. In addition, with the development of the Chinese economy, the

structure of factor endowments in China has undergone significant changes, and advanced production factors, such as capital and technology, have accumulated to a certain extent. CMEs have gradually adopted targeted OFDI modes in host countries according to their investment motivation needs and enterprise development stages. From the perspective of resource-seeking motivation, in columns (5)–(6), it can be seen that the interaction term *idiff*res* between institutional differences and resource-seeking motivation was positive but not significant in column (5) and positive at the 10% level of significance (See the interaction term *idiff*res* between institutional differences and market-seeking motivation in column (6) after controlling for variables. Institutional differences significantly promote the M&A performance of CMEs with respect to their resource-seeking motivation, indicating that institutional differences have met the investment preference needs of CMEs and promoted their M&A in the host country. Therefore, Hypothesis 4 was validated.

**Table 6.** Regression results of the moderating effects of different investment motivations.

|  | (1) | (2) | (3) | (4) | (5) | (6) |
|---|---|---|---|---|---|---|
| *idiff* | 1.092 *** | 1.084 *** | 1.184 *** | 1.144 *** | 1.168 *** | 1.169 *** |
|  | (0.089) | (0.095) | (0.084) | (0.090) | (0.078) | (0.083) |
| *lngdp* | 0.107 *** | 0.060 |  |  |  |  |
|  | (0.040) | (0.043) |  |  |  |  |
| *idiff*lngdp* | −0.032 | −0.100 |  |  |  |  |
|  | (0.061) | (0.066) |  |  |  |  |
| *tec* |  |  | −0.012 *** | −0.005 |  |  |
|  |  |  | (0.004) | (0.005) |  |  |
| *idiff*tec* |  |  | −0.020 *** | −0.023 *** |  |  |
|  |  |  | (0.006) | (0.006) |  |  |
| *res* |  |  |  |  | −0.006 | −0.016 |
|  |  |  |  |  | (0.010) | (0.011) |
| *idiff*res* |  |  |  |  | 0.014 | 0.028 * |
|  |  |  |  |  | (0.015) | (0.017) |
| *lnstaff* |  | −0.253 *** |  | −0.238 *** |  | −0.240 *** |
|  |  | (0.085) |  | (0.085) |  | (0.085) |
| *lnfa* |  | 0.210 *** |  | 0.197 *** |  | 0.201 *** |
|  |  | (0.070) |  | (0.070) |  | (0.070) |
| *cap* |  | 0.126 ** |  | 0.138 ** |  | 0.128 ** |
|  |  | (0.056) |  | (0.058) |  | (0.057) |
| *age* |  | 0.022 ** |  | 0.024 ** |  | 0.023 ** |
|  |  | (0.011) |  | (0.011) |  | (0.011) |
| *lngfcf* |  | −0.789 ** |  | −0.751 * |  | −0.943 ** |
|  |  | (0.380) |  | (0.399) |  | (0.403) |
| *lndis* |  | 0.259 ** |  | 0.224 ** |  | 0.289 *** |
|  |  | (0.112) |  | (0.104) |  | (0.101) |
| Constant | −4.091 *** | −5.265 ** | −0.994 * | −3.224 | −1.243 ** | −3.375 * |
|  | (1.161) | (2.152) | (0.514) | (2.068) | (0.501) | (2.040) |
| Province FE | Yes | Yes | Yes | Yes | Yes | Yes |
| Year FE | Yes | Yes | Yes | Yes | Yes | Yes |
| Observations | 1688 | 1688 | 1692 | 1692 | 1692 | 1692 |
| Pseudo R$^2$ | 0.159 | 0.180 | 0.168 | 0.187 | 0.158 | 0.182 |
| Log Lik | −968.3 | −944.3 | −960.2 | −937.2 | −971.6 | −943.3 |
| LR Chi$^2$ | 297.9 | 310.4 | 298.6 | 318.3 | 294.5 | 322.0 |
| Prob > chi$^2$ | 0.000 | 0.000 | 0.000 | 0.000 | 0.000 | 0.000 |

Note: *, **, and *** represent significance at the 10%, 5%, and 1% levels, respectively; the robust standard errors are in parentheses.

The moderating effect of the "Belt and Road" Initiative. To examine the moderating effect of the "Belt and Road" Initiative on the relationship between institutional differences and the OFDI entry mode of CMEs, we constructed the interaction term *idiff*cd* between the institutional differences and the signing of "Belt and Road" cooperation agreements. As

shown in the regression results in Table 7, column (1) indicates that the regression coefficient of the interaction term *idiff*cd* between institutional differences and the signing of "Belt and Road" cooperation agreements was positive but significant only at the 10% level when no control variables were included. Column (2) and (3) show that the regression coefficient of the interaction term *idiff*cd* was significant at the 5% level after controlling for province and time fixed-effects, indicating that the influence of other factors had been effectively eliminated. Column (4) shows that, after controlling for both the control variables and the province and time fixed-effects, the interaction term coefficient was 0.559 and remained significant at the 5% level. This suggests that the signing of "Belt and Road" cooperation agreements has a positive moderating effect on the relationship between institutional differences and M&A by CMEs. In other words, the greater the institutional differences, the more likely that CMEs that have signed "Belt and Road" cooperation agreements will choose M&A as their investment strategy in the host country.

**Table 7.** Regression results of the moderating effect of the BRI.

| | (1) | (2) | (3) | (4) |
|---|---|---|---|---|
| *idiff* | 1.142 *** | 1.192 *** | 1.154 *** | 1.181 *** |
| | (0.083) | (0.090) | (0.086) | (0.090) |
| *cd* | 0.011 | 0.364 | 0.328 | 0.336 |
| | (0.222) | (0.238) | (0.243) | (0.245) |
| *idiff*cd* | 0.468 * | 0.549 ** | 0.572 ** | 0.559 ** |
| | (0.246) | (0.258) | (0.253) | (0.256) |
| *lnstaff* | | −0.253 *** | −0.255 *** | −0.246 *** |
| | | (0.084) | (0.078) | (0.085) |
| *lnfa* | | 0.210 *** | 0.194 *** | 0.208 *** |
| | | (0.069) | (0.066) | (0.070) |
| *cap* | | 0.146 *** | 0.090 * | 0.128 ** |
| | | (0.056) | (0.050) | (0.057) |
| *age* | | 0.026 *** | 0.019 * | 0.023 ** |
| | | (0.010) | (0.010) | (0.011) |
| *lngfcf* | | −0.951 ** | −0.960 *** | −0.911 ** |
| | | (0.376) | (0.361) | (0.379) |
| *lndis* | | 0.278 *** | 0.259 *** | 0.264 *** |
| | | (0.096) | (0.095) | (0.097) |
| Constant | −1.209 ** | −3.888 ** | −3.184 * | −3.431 * |
| | (0.483) | (1.892) | (1.835) | (1.956) |
| Province FE | Yes | Yes | No | Yes |
| Year FE | Yes | No | Yes | Yes |
| Observations | 1692 | 1692 | 1692 | 1692 |
| Pseudo $R^2$ | 0.159 | 0.176 | 0.157 | 0.182 |
| Log Lik | −970.2 | −950.3 | −972.0 | −943.2 |
| LR $Chi^2$ | 284.9 | 290.7 | 278.0 | 309.0 |
| Prob > $chi^2$ | 0.000 | 0.000 | 0.000 | 0.000 |

Note: *, **, and *** represent significance at the 10%, 5%, and 1% levels, respectively; the robust standard errors are in parentheses.

Moreover, this finding supports the conclusion that the "Belt and Road" Initiative can effectively promote Chinese M&A activities and has a complementary optimizing effect on institutional differences (Hypothesis 5). In recent years, some industries have faced the test of chain-breaking or shutdown due to global political and economic turmoil, the sudden and widespread outbreak of the COVID-19 pandemic, and the strong impact on global industrial and supply chains. Countries have come to pay more attention to the protection of critical industries and core technologies, leading some Chinese multi-national corporations to become hesitant to engage in overseas M&A activities in the short-term. The signing of "Belt and Road" cooperation agreements strengthens political mutual trust between China and the host country and effectively reduces cooperation risks, providing

protection for implementing and further promoting Chinese overseas investment and cooperation projects.

### 4.5. Heterogeneity Test

Analysis of ownership structure. As shown in the regression results in Table 8, in the regression results for the sub-samples, columns (1) and (2) represent the samples of state-owned and non-state-owned CMEs, respectively. Columns (1)–(2) show that, when Chinese outward direct investment enterprises are state-owned manufacturing enterprises, the coefficient of the interaction term *idiff\*cd* between institutional differences and the signing of the "Belt and Road" cooperation agreement was positive and significant at the 5% level. In particular, the interaction term coefficient for state-owned manufacturing enterprises was significantly higher than that for non-state-owned manufacturing enterprises. The impact of institutional differences on the choice of M&A by state-owned manufacturing enterprises was more prominent in countries that have signed the "Belt and Road" cooperation agreement. This suggests that signing a "Belt and Road" cooperation agreement under institutional differences increases the likelihood that state-owned manufacturing enterprises choose M&A. Chinese state-owned manufacturing enterprises are pioneers in the "Belt and Road" construction, acting as the main force and playing a leading role. Most "Belt and Road" cooperation involves significant infrastructure projects. Compared with non-state-owned manufacturing enterprises, state-owned manufacturing enterprises possess scale, technology, equipment, and competitiveness advantages. They are more capable of undertaking major projects, thus playing an essential role in promoting the international cooperation process of production capacity and manufacturing. The related demonstration effect is gradually emerging.

**Table 8.** Heterogeneity regression results of the BRI.

|  | (1) | (2) | (3) | (4) |
|---|---|---|---|---|
|  | **SOE** | **Non-SOE** | **Asian** | **Non-Asian** |
| *idiff* | 1.515 *** | 1.061 *** | 0.847 *** | 1.329 *** |
|  | (0.177) | (0.112) | (0.176) | (0.143) |
| *cd* | 0.779 | 0.384 | 1.057 *** | −0.106 |
|  | (0.542) | (0.280) | (0.365) | (0.355) |
| *idiff\*cd* | 1.335 ** | 0.472 | 0.399 | 1.197 *** |
|  | (0.587) | (0.308) | (0.376) | (0.450) |
| *lnstaff* | −0.559 *** | −0.160 | −0.068 | −0.375 *** |
|  | (0.170) | (0.111) | (0.145) | (0.117) |
| *lnfa* | 0.348 ** | 0.180 * | 0.202 * | 0.249 ** |
|  | (0.140) | (0.095) | (0.115) | (0.097) |
| *cap* | −0.165 | 0.133 ** | 0.088 | 0.175 ** |
|  | (0.163) | (0.062) | (0.094) | (0.078) |
| *age* | 0.060 ** | 0.016 | 0.041 ** | 0.013 |
|  | (0.030) | (0.013) | (0.020) | (0.014) |
| *lngfcf* | −1.487 * | −0.661 | −0.178 | −1.655 *** |
|  | (0.768) | (0.477) | (0.806) | (0.603) |
| *lndis* | −0.038 | 0.354 *** | −0.236 | −1.334 *** |
|  | (0.231) | (0.114) | (0.245) | (0.332) |
| Constant | 0.397 | −5.050 ** | −2.411 | 13.104 *** |
|  | (4.238) | (2.465) | (3.697) | (3.865) |
| Province FE | Yes | Yes | Yes | Yes |
| Year FE | Yes | Yes | Yes | Yes |
| Observations | 478 | 1171 | 593 | 1065 |
| Pseudo $R^2$ | 0.318 | 0.146 | 0.127 | 0.225 |
| Log Lik | −218.2 | −686.9 | −313.2 | −571.7 |
| LR $Chi^2$ | 136.6 | 188.8 | 81.89 | 203.2 |
| Prob > $chi^2$ | 0.000 | 0.000 | 0.000 | 0.000 |

Note: *, **, and *** represent significance at the 10%, 5%, and 1% levels, respectively; the robust standard errors are in parentheses.

Analysis of national regional levels. Columns (3) and (4) represent the sub-samples of OFDI destination countries in Asia and non-Asia, respectively. From the regression results of the sub-samples, it can be seen that, in non-Asian countries, the coefficient of the

interaction term *idiff*cd* was significant at the 1% level and was positive. This means that if a host country is non-Asian and signs the "Belt and Road" cooperation agreement with China, it will promote the tendency of CMEs to choose M&A. This suggests that signing the "Belt and Road" cooperation agreement can effectively reduce the enormous pressure and risks faced by CMEs when conducting M&A caused by institutional differences, thus enhancing their confidence in "going out."

### 4.6. Robustness Test

Model transformation method. In order to ensure the accuracy and reliability of the conclusions, robustness checks were conducted in this study. On the one hand, the probit model was used to regress the research hypotheses, according to columns (1)–(4) as shown in Table 9, by replacing the econometric method. On the other hand, the robustness of the moderating effect of the "Belt and Road" Initiative was also tested using the probit model, and the empirical results from columns (1)–(4) are shown in Table 10. The results re-estimated using the probit model were consistent with those estimated using the baseline regression, which confirmed the hypotheses proposed earlier.

**Table 9.** Regression results of the probit model on the basic regression.

|  | (1) | (2) | (3) | (4) |
|---|---|---|---|---|
| *idiff* | 0.701 *** | 0.697 *** | 0.687 *** | 0.696 *** |
|  | (0.044) | (0.047) | (0.046) | (0.048) |
| *lnstaff* |  | −0.157 *** | −0.158 *** | −0.150 *** |
|  |  | (0.049) | (0.046) | (0.050) |
| *lnfa* |  | 0.122 *** | 0.115 *** | 0.120 *** |
|  |  | (0.041) | (0.039) | (0.041) |
| *cap* |  | 0.083 *** | 0.051 * | 0.073 ** |
|  |  | (0.032) | (0.029) | (0.032) |
| *age* |  | 0.016 *** | 0.011 * | 0.014 ** |
|  |  | (0.006) | (0.006) | (0.006) |
| *lngfcf* |  | −0.446 ** | −0.478 ** | −0.432 ** |
|  |  | (0.211) | (0.205) | (0.212) |
| *lndis* |  | 0.170 *** | 0.157 *** | 0.162 *** |
|  |  | (0.057) | (0.057) | (0.057) |
| Constant | −0.760 *** | −2.586 ** | −2.147 ** | −2.303 ** |
|  | (0.290) | (1.098) | (1.078) | (1.134) |
| Province FE | Yes | Yes | No | Yes |
| Year FE | Yes | No | Yes | Yes |
| Observations | 1692 | 1692 | 1692 | 1692 |
| Pseudo $R^2$ | 0.158 | 0.174 | 0.156 | 0.180 |
| Log Lik | −971.3 | −952.8 | −974.0 | −945.4 |
| LR Chi$^2$ | 334.9 | 343.7 | 328.1 | 367.5 |
| Prob > chi$^2$ | 0.000 | 0.000 | 0.000 | 0.000 |

Note: *, **, and *** represent significance at the 10%, 5%, and 1% levels, respectively; the robust standard errors are in parentheses.

**Table 10.** Regression results of the probit model on the moderating effect of the BRI.

|  | (1) | (2) | (3) | (4) |
|---|---|---|---|---|
| *idiff* | 0.686 *** | 0.707 *** | 0.692 *** | 0.702 *** |
|  | (0.047) | (0.051) | (0.049) | (0.051) |
| *cd* | 0.013 | 0.216 | 0.201 | 0.207 |
|  | (0.133) | (0.137) | (0.141) | (0.142) |
| *idiff*cd* | 0.247 * | 0.299 ** | 0.309 ** | 0.305 ** |
|  | (0.136) | (0.141) | (0.137) | (0.140) |
| *lnstaff* |  | −0.156 *** | −0.158 *** | −0.151 *** |
|  |  | (0.049) | (0.046) | (0.050) |

**Table 10.** *Cont.*

|  | **(1)** | **(2)** | **(3)** | **(4)** |
|---|---|---|---|---|
| *lnfa* |  | 0.125 *** | 0.117 *** | 0.123 *** |
|  |  | (0.041) | (0.039) | (0.041) |
| *cap* |  | 0.085 *** | 0.052 * | 0.074 ** |
|  |  | (0.032) | (0.029) | (0.032) |
| *age* |  | 0.016 *** | 0.011 * | 0.014 ** |
|  |  | (0.006) | (0.006) | (0.006) |
| *lngfcf* |  | −0.515 ** | −0.543 *** | −0.501 ** |
|  |  | (0.215) | (0.207) | (0.216) |
| *lndis* |  | 0.170 *** | 0.156 *** | 0.161 *** |
|  |  | (0.057) | (0.057) | (0.058) |
| Constant | −0.727 ** | −2.443 ** | −1.985 * | −2.147 * |
|  | (0.287) | (1.105) | (1.083) | (1.140) |
| Province FE | Yes | Yes | No | Yes |
| Year FE | Yes | No | Yes | Yes |
| Observations | 1692 | 1692 | 1692 | 1692 |
| Pseudo $R^2$ | 0.159 | 0.176 | 0.158 | 0.182 |
| Log Lik | −969.4 | −950.3 | −971.3 | −943.0 |
| LR $Chi^2$ | 326.3 | 335.7 | 320.7 | 360.6 |
| Prob > $chi^2$ | 0.000 | 0.000 | 0.000 | 0.000 |

Note: *, **, and *** represent significance at the 10%, 5%, and 1% levels, respectively; the robust standard errors are in parentheses.

Replacement of independent variables. We used the "rule of law" (*rl*) to measure institutional differences, which further confirmed the significant impact of institutional differences on the OFDI entry mode choice of CMEs. In Table 11, columns (1)–(4) show the results after replacing the independent variable in the basic regression, which were consistent with the basic regression results and were positive at the 1% significance level. In Table 12, columns (1)–(4) show the results after replacing the independent variable and interaction terms of the "Belt and Road" Initiative. Compared with before the replacement, the coefficient of the rule of law and the interaction term of signing the "Belt and Road" cooperation agreement was significantly positive at the 1% level, indicating that the rule of law—as the institutional basis of a harmonious society—is not only a guarantee of social fairness and national stability but also a factor that CMEs must pay attention to when making OFDI decisions. The better the legal environment of the host country, the more conducive it is to investment cooperation under the "Belt and Road" framework. Therefore, the research conclusions of this paper can be considered robust and reliable, and the research hypotheses were effectively validated.

**Table 11.** Regression results of replacement of independent variable (1).

|  | **(1)** | **(2)** | **(3)** | **(4)** |
|---|---|---|---|---|
| *rl* | 0.976 *** | 0.992 *** | 0.957 *** | 0.981 *** |
|  | (0.068) | (0.071) | (0.068) | (0.072) |
| *lnstaff* |  | −0.254 *** | −0.261 *** | −0.250 *** |
|  |  | (0.084) | (0.078) | (0.084) |
| *lnfa* |  | 0.202 *** | 0.188 *** | 0.200 *** |
|  |  | (0.068) | (0.065) | (0.069) |
| *cap* |  | 0.138 ** | 0.086 * | 0.126 ** |
|  |  | (0.055) | (0.050) | (0.056) |
| *age* |  | 0.020 ** | 0.017 | 0.021 * |
|  |  | (0.010) | (0.010) | (0.011) |
| *lngfcf* |  | −0.902 ** | −0.937 *** | −0.863 ** |
|  |  | (0.355) | (0.343) | (0.359) |

**Table 11.** *Cont.*

|  | **(1)** | **(2)** | **(3)** | **(4)** |
|---|---|---|---|---|
| *lndis* |  | 0.307 *** | 0.289 *** | 0.298 *** |
|  |  | (0.095) | (0.094) | (0.096) |
| Constant | −0.538 | −3.507 * | −2.622 | −2.928 |
|  | (0.470) | (1.823) | (1.773) | (1.886) |
| Province FE | Yes | Yes | No | Yes |
| Year FE | Yes | No | Yes | Yes |
| Observations | 1692 | 1692 | 1692 | 1692 |
| Pseudo $R^2$ | 0.147 | 0.167 | 0.147 | 0.173 |
| Log Lik | −983.5 | −960.9 | −983.7 | −954.3 |
| LR $Chi^2$ | 282.4 | 302.2 | 290.5 | 320.7 |
| Prob > $chi^2$ | 0.000 | 0.000 | 0.000 | 0.000 |

Note: *, **, and *** represent significance at the 10%, 5%, and 1% levels, respectively; the robust standard errors are in parentheses.

**Table 12.** Regression results of the replacement of the independent variable (2).

|  | **(1)** | **(2)** | **(3)** | **(4)** |
|---|---|---|---|---|
| *rl* | 0.955 *** | 1.007 *** | 0.975 *** | 0.999 *** |
|  | (0.072) | (0.077) | (0.075) | (0.078) |
| *cd* | −0.076 | 0.257 | 0.297 | 0.300 |
|  | (0.216) | (0.235) | (0.240) | (0.241) |
| *rl\*cd* | 0.591 ** | 0.727 *** | 0.757 *** | 0.747 *** |
|  | (0.236) | (0.256) | (0.251) | (0.254) |
| *lnstaff* |  | −0.253 *** | −0.257 *** | −0.248 *** |
|  |  | (0.083) | (0.078) | (0.084) |
| *lnfa* |  | 0.208 *** | 0.191 *** | 0.206 *** |
|  |  | (0.069) | (0.066) | (0.069) |
| *cap* |  | 0.140 ** | 0.090 * | 0.128 ** |
|  |  | (0.055) | (0.050) | (0.056) |
| *age* |  | 0.021 ** | 0.018 * | 0.022 ** |
|  |  | (0.010) | (0.010) | (0.011) |
| *lngfcf* |  | −1.043 *** | −1.078 *** | −1.022 *** |
|  |  | (0.362) | (0.346) | (0.364) |
| *lndis* |  | 0.295 *** | 0.281 *** | 0.289 *** |
|  |  | (0.095) | (0.094) | (0.096) |
| Constant | −0.512 | −3.121 * | −2.234 | −2.516 |
|  | (0.458) | (1.832) | (1.778) | (1.896) |
| Province FE | Yes | Yes | No | Yes |
| Year FE | Yes | No | Yes | Yes |
| Observations | 1692 | 1692 | 1692 | 1692 |
| Pseudo $R^2$ | 0.151 | 0.171 | 0.152 | 0.177 |
| Log Lik | −979.1 | −956.1 | −978.2 | −949.2 |
| LR $Chi^2$ | 273.1 | 289.7 | 276.3 | 307.6 |
| Prob > $chi^2$ | 0.000 | 0.000 | 0.000 | 0.000 |

Note: *, **, and *** represent significance at the 10%, 5%, and 1% levels, respectively; the robust standard errors are in parentheses.

### 4.7. Endogeneity Test

Omitted variable bias and bidirectional causality are the main reasons for endogeneity problems. Therefore, we conducted practical endogeneity tests to alleviate the impact of endogeneity problems on the estimation results as much as possible. OFDI is a strategic decision of an enterprise and belongs to micro-subject behavior, while national institutions are the basic national policy; as the behavior of individual enterprises cannot easily influence the national will, the possibility of the OFDI behaviors of enterprises inversely affecting national institutions is relatively weak. Therefore, the problem of bidirectional causality in this paper was relatively weak. While considering enterprise-level variables that affect the OFDI entry mode of CMEs (e.g., enterprise size, capital intensity, and enterprise age),

as well as macro variables (e.g., host country infrastructure and geographical distance), we also controlled for province and time fixed-effects to alleviate the problem of omitted variables leading to endogeneity.

In particular, we consider the endogeneity problem caused by potential omitted variable bias, which leads to biased estimation results. Therefore, we took political affinity as an instrumental variable in 2SLS regression using United Nations General Assembly voting data. The United Nations General Assembly voting data are highly correlated with national institutions and the institutional differences between countries, which is exogenous with respect to the investment mode decisions of Chinese enterprises. Table 13 shows that, in the first stage, there was a significant positive correlation between political affinity and institutional differences in columns (1) and (3), with both at the 1% significance level. In the second-stage regression column (2), the coefficient of institutional differences and the coefficient of the interaction term *idiff\*cd* in the column (4) were both significantly positive at the 1% significance level. The regression results demonstrate that, when using the political affinity index based on United Nations General Assembly voting data as an instrumental variable, the promoting effect of institutional differences on the choice of M&A by CMEs was still robust. Furthermore, under the influence of institutional differences, signing a "Belt and Road" cooperation agreement still positively moderated the choice of M&A by CMEs.

**Table 13.** Instrumental variables regression.

| | (1) | (2) | (3) | (4) |
|---|---|---|---|---|
| | **First Stage** | **Second Stage** | **First Stage** | **Second Stage** |
| *idiff* | | 0.930 *** | | 0.964 *** |
| | | (0.081) | | (0.085) |
| *ipd* | 0.433 *** | | 0.408 *** | |
| | (0.019) | | (0.018) | |
| *cd* | | | −0.664 *** | 0.418 *** |
| | | | (0.063) | (0.149) |
| *idiff\*cd* | | | 0.175 *** | 0.274 * |
| | | | (0.056) | (0.142) |
| *lnstaff* | −0.033 | −0.136 *** | −0.035 | −0.134 *** |
| | (0.024) | (0.050) | (0.022) | (0.050) |
| *lnfa* | −0.037 * | 0.131 *** | −0.022 | 0.131 *** |
| | (0.020) | (0.041) | (0.019) | (0.041) |
| *cap* | 0.019 | 0.066 ** | 0.010 | 0.069 ** |
| | (0.015) | (0.031) | (0.014) | (0.031) |
| *age* | −0.015 *** | 0.018 *** | −0.013 *** | 0.018 *** |
| | (0.003) | (0.006) | (0.003) | (0.006) |
| *lngfcf* | −0.580 *** | −0.229 | −0.479 *** | −0.313 |
| | (0.093) | (0.223) | (0.087) | (0.224) |
| *lndis* | −0.422 *** | 0.162 *** | −0.465 *** | 0.181 *** |
| | (0.032) | (0.056) | (0.030) | (0.057) |
| Constant | 7.662 *** | −3.700 *** | 7.425 *** | −3.652 *** |
| | (0.513) | (1.194) | (0.478) | (1.203) |
| Province FE | Yes | Yes | Yes | Yes |
| Year FE | Yes | Yes | Yes | Yes |
| Sigma$^2$ | | −0.396 *** | | −0.471 *** |
| | | (0.017) | | (0.017) |
| Log Lik | | −2671 | | −2542 |
| LR Chi$^2$ | | 316.7 | | 331.0 |
| Observations | 1692 | 1692 | 1692 | 1692 |
| Wald-test | | 10.80 | | 11.93 |
| | | [0.000] | | [0.000] |

Note: *, **, and *** represent significance at the 10%, 5%, and 1% levels, respectively; the robust standard errors are in parentheses; the *p*-value of the Wald test is in square brackets.

## 5. Conclusions and Policy Recommendations

For this article, we took 1692 OFDI events of 735 A-share listed companies in China's manufacturing industry from 2010 to 2019 as research samples. Using the logit model, we examined the influence of institutional differences between China and host countries on the OFDI entry mode choice of Chinese enterprises. Furthermore, we explored the moderating effects of investment motivations and the "Belt and Road" Initiative on institutional differences.

The results of the study led to the following conclusions: First, the greater the institutional differences between the host country and China, the more likely that CMEs will tend to choose M&A. While exploring the impact of the six dimensions of institutional factors, it was found that voice and accountability, political stability, government effectiveness, regulatory quality, rule of law, and control of corruption can all positively promote the choice of M&A by CMEs. Second, different investment motivations have different moderating effects on the institutional differences and the OFDI entry mode choice of CMEs. Based on resource-seeking motivations, institutional differences are positively correlated with the M&A choice of CMEs. Resource-seeking motivations have a positive moderating effect on the choice of M&A by CMEs, while technology-seeking motivations have a negative moderating effect on this choice. Third, the signing of cooperation documents under the "Belt and Road" Initiative has a positive moderating effect on the relationship between institutional differences and choice of M&A by CMEs; that is, the greater the institutional differences, the more that CMEs prefer to choose M&A for investment in destination countries that have signed the cooperation documents. By distinguishing between corporate ownership, under the influence of institutional differences, signing the "Belt and Road" cooperation documents has dramatically promoted the choice of M&A by Chinese state-owned manufacturing enterprises compared with non-state-owned enterprises. At the same time, if the host country is a non-Asian country and signs a "Belt and Road" cooperation document with China, it will be conducive to promoting the choice of M&A by CMEs.

The policy implications of this paper suggest that the entry mode choice of an enterprise is an important strategic decision. Under the influence of institutional differences between the home country and the host country, M&A or greenfield investments should be designed and arranged based on the institutional advantages of different target countries and the different investment motivations of multi-national enterprises. The measures taken by the host country to improve political stability and government efficiency will help promote Chinese enterprises to actively engage in OFDI. The BRI is crucial for China to strengthen economic cooperation and mutual trust with various countries and regions worldwide. Adhering to the acceleration of the "Belt and Road" construction, investment and cooperative relationships with "Belt and Road" partners will be further deepened, providing a good external environment for Chinese enterprises to engage in OFDI, especially with respect to the broad prospects of greenfield investment within the "Belt and Road" region. Considering various stakeholders specifically, we can consider the following aspects.

For government entities, in the process of OFDI, attention should be paid to the institutional differences between countries, particularly with respect to institutional arrangements such as the laws and regulations of the host country. This will help to actively promote institutional co-construction, enhance institutional mutual trust, break down institutional barriers, and enhance benign interaction with international institutions. The advantages related to policy preference and resource integration should be relied upon in order to help countries in the BRI region establish a preferential policy framework and provide support and guarantee for common development; for example, the construction and improvement of legal systems among countries in the BRI region may be strengthened, ensuring the effective implementation of contracts and the protection of intellectual property rights. Furthermore, it is important to focus on grasping institutional factors and providing effective decision-making guidance; for example, when investing in host countries with significant institutional differences, CMEs should be guided to prioritize M&A. It is critical to actively

advocate and mobilize more countries, regions, and international organizations to sign BRI cooperation documents, carry out bilateral friendly state visits within the region, remedy political differences, strengthen political mutual trust, further promote international economic cooperation, and inject new impetus into the field of BRI cooperation.

Regarding enterprise entities, CMEs should adjust their foreign investment strategies according to their practical requirements; make reasonable decisions on investment modes; accurately assess the impact of the institutional environment based on a complete understanding of the host country's local political and legal constraints, as well as the differences in institutional arrangements between the host country and the home country; avoid large-scale investments in countries and regions where wars are frequent and political situations are unstable; establish and strengthen investment risk prevention and control mechanisms; and reduce the institutional risks of M&A or greenfield investments. Under the influence of institutional differences, the greater the institutional differences, the more that CMEs based on resource-seeking motivations should prioritize M&A. Those based on technology-seeking motivations can consider entering the host country market through greenfield investments in order to reduce investment risks. Considering the promotion effect of the BRI, CMEs can give priority to the host countries or regions that have signed cooperation documents when investing overseas. While actively responding to national policies, Chinese enterprises should also face the differences in the choice of OFDI entry mode depending on the target country, partner, product, talent, and service. Through the deep integration of the construction of the BRI with international and regional cooperation platforms, enterprises may continuously improve and upgrade their existing products and technologies, enhance international competitiveness, comply with and guide overseas market demands, and strive to break through the encirclement of overseas competitive enterprises. Advantageous factor endowments and comparative advantage industries of countries with different systems should be identified, the optimization of resource allocation between regions and the integration and division of labor between markets should be strengthened, the specific advantages of China's manufacturing industry in infrastructure construction and other aspects should be leveraged, the optimization of China's manufacturing industry layout in high-end manufacturing industries abroad should be accelerated, and the development of manufacturing industry clusters should be promoted. Under the leadership of state-owned manufacturing enterprises, we will gradually pay attention to and strongly support the external investment of private manufacturing enterprises in China. With the help of industry associations, industry alliances, research institutes, and other organizational structures, multiple resources and enterprises can be coordinated to solve the problem of private manufacturing enterprises going global, ensuring the smooth export channel of high-quality production capacity.

The principal theoretical implication of this study is that the research results demonstrate the uniqueness in the choice of OFDI entry mode by Chinese enterprises in the context of the new era; additionally, the study indicates the varying impacts of institutional differences on the implementation of M&A and greenfield investment by enterprises, the expanded understanding of the institutional environment in emerging countries in the field of international business, and the promotion of the application and development of institutional theory in the field of international business. At the same time, the research results indicate the OFDI behavior preference of Chinese enterprises under the "Belt and Road" construction, which can effectively supplement the theoretical literature related to "Belt and Road" construction and help to promote theoretical innovation and system construction in the context of the "Belt and Road".

This study is not without limitations, and future work may extend our research in multiple ways. First, this paper comprises an exploratory attempt to study how differentiated institutional environments work in conjunction with government policies to influence the OFDI entry mode choice of enterprises. As such, future research could explore and clarify the factors influencing the OFDI entry mode through the consideration of mediating variables. Second, in the selection of samples, we primarily focused on manufacturing

enterprises as the research object. In future research, multiple industries can be added to the sample for comparative analysis, which may lead to more persuasive research findings. In addition, the "Belt and Road" Initiative positively affects overseas investment by enterprises. In future research, the effects of the initiative can be refined in terms of the influence mechanism by sorting out changes in enterprise investment behavior. In general, further developments and trends of OFDI mode research can be explored from multiple perspectives. For instance, we can further evaluate the choice of OFDI mode under different investment motivations or how the institutional environment affects the OFDI entry mode choice regarding export-platform OFDI and other investment types. It is also possible to discuss the OFDI entry mode choice of Chinese enterprises under uncertain conditions or other situations.

**Author Contributions:** Conceptualization, Q.X. and H.Y.; methodology, Q.X. and H.Y.; formal analysis, Q.X. and H.Y.; writing—original draft, Q.X.; funding acquisition, H.Y. All authors have read and agreed to the published version of the manuscript.

**Funding:** This work was jointly supported by the National Office for Philosophy and Social Sciences of China (grant number 16BJY079) and the Philosophy and Social Sciences Foundation of Hunan Province (grant number 22JD002).

**Institutional Review Board Statement:** Not applicable.

**Informed Consent Statement:** Not applicable.

**Data Availability Statement:** The data are available from the corresponding author upon request.

**Conflicts of Interest:** The authors declare no conflict of interest.

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
