# Peer review of "Institutional Differences and the Choice of Outward Foreign Direct Investment Mode under the “Belt and Road” Initiative: Experience Analysis Based on China’s Manufacturing Enterprises"

_sustainability, doi:10.3390/su15097201_

Round 1
Reviewer 1 Report
Dear author(s), Thank you very much for submitting your valuable work to sustainability. I read this article title as “Institutional Difference and the Choice of Outward Foreign Direct Investment Mode under the "Belt and Road" Initiative: Experience analysis based on China's manufacturing enterprises”. This study investigates the Institutional Difference and the Choice of Outward Foreign Direct Investment Mode under the "Belt and Road Initiative" (BRI) by using China's manufacturing enterprises (CMEs) on Logit mode. However, in some lines and sentences there are still minor issues. I am requesting the authors to have a relook and correct it.
1. Abstract (Minor)
The overall abstract is looking great but still it has some minor issue such as "Belt and Road Initiative" (BRI) in the abstract like this.
If Possible please revise this sentence it is too general, and add the a sentences that has both practical and theoretical contribution. “This study would contribute to promoting institutional construction and strengthen international cooperation in the "Belt and Road", further effectively promoting the high-quality development of China's manufacturing industry”.
2. Introduction (Perfect). The novelty is not too clear, I am suggesting you to read this articles.
https://www.mdpi.com/2071-1050/11/24/7055 .
Another suggestion is that please concise your contribution and novelty in the introduction section.
3. Literature review (Minor). The literature review of this paper is slightly good and well written. However, if you have a look of this section from 37-50 only three citations are used from the current literature i.e. 2019, 2020, and 2022. I suggest to support your literature with the current update literature. Some important papers are more helpful for your work. Please spend sometime to read and cite.
https://doi.org/10.1016/j.qref.2020.12.001
https://doi.org/10.1080/14631377.2020.1745560
https://doi.org/10.1002/ise3.15
4. Last but not the least, it is suggested to proofread this publishable article that will improve its language and rectify other typos mistakes.
Reviewer 2 Report
I suggest several comments to improve this work. First) I suggest updating the literature review by adding recently published studies such as: A) Global competitiveness and capital flows: does stage of economic development and risk rating matter?. Asia-Pacific Journal of Accounting & Economics, 2020, 27(4), 426-450. B) Nexus between institutional quality and capital inflows at different stages of economic development. International Review of Finance, 2019, 19(2), 435-445. Second) The descriptive results should be improved as follows: 1) It should be added the Granger causality. 2) It should be added the unit root test with structural breaks. 3) The statistical significance with 1% should be shown by a star in Table 3. 4) It should be added the graphs of variables to show the trends over time. Third) the policy implications should be explained to the various stakeholders both theoretically and practically.
Reviewer 3 Report
1. What is the main question addressed by the research?
This study examines outward foreign direct investment motivation and the impact of institutional differences between China and the host country.
2. Do you consider the topic original or relevant in the field? Does it
address a specific gap in the field?
I think the topic is relevant. This study clearly stated the gap of the study and also emphasized the different between this study and other studies.
3. What does it add to the subject area compared with other published
material?
This study contributed to the subject area in terms of incorporating the impact of institutional differences on Chinese enterprises' OFDI mode choices of the Belt and Road initiative's geopolitical pattern and international cooperation.
This study also explored the moderation effect of the OFDI mode choices. Another contribution of this study is that the introduction of the political affinity index as a tool variable to identify the impact of institutional differences on Chinese enterprises' OFDI mode choices.
4. What specific improvements should the authors consider regarding the
methodology? What further controls should be considered?
I think the methodology is appropriate.
5. Are the conclusions consistent with the evidence and arguments presented
and do they address the main question posed?
Yes. The conclusion is consistent with the discussion of the results.
6. Are the references appropriate?
The references are adequate.
7. Please include any additional comments on the tables and figures.
The tables and figures are adequate.
Reviewer 4 Report
It was a pleasure to rad the paper. It is very well structured and developed. congratulations to the authors for this analysis.
One minor suggestion - theoretical implication and further developments to better highlighted
Round 2
Reviewer 2 Report
The authors addressed my comments and the paper has the potential for publishing in the current format.